# Abscisic Acid Metabolizing *Rhodococcus* sp. Counteracts Phytopathogenic Effects of Abscisic Acid Producing *Botrytis* sp. on Sunflower Seedlings

**DOI:** 10.3390/plants14152442

**Published:** 2025-08-07

**Authors:** Alexander I. Shaposhnikov, Oleg S. Yuzikhin, Tatiana S. Azarova, Edgar A. Sekste, Anna L. Sazanova, Nadezhda A. Vishnevskaya, Vlada Y. Shahnazarova, Polina V. Guro, Miroslav I. Lebedinskii, Vera I. Safronova, Yuri V. Gogolev, Andrey A. Belimov

**Affiliations:** 1All-Russia Research Institute for Agricultural Microbiology, Saint-Petersburg 196608, Russia; ai.shaposhnikov@arriam.ru (A.I.S.); os.yuzikhin@arriam.ru (O.S.Y.); sekste_edgar@mail.ru (E.A.S.); navishnevskaya@rambler.ru (N.A.V.); vi.safronova@arriam.ru (V.I.S.); 2Kazan Institute of Biochemistry and Biophysics of the Federal Research Center “Kazan Scientific Center of the RAS”, Kazan 420111, Russia; gogolev@kibb.knc.ru

**Keywords:** abscisic acid, biocontrol, *Botrytis*, phytohormones, PGPR, phytopathogens

## Abstract

One of the important traits of many plant growth-promoting rhizobacteria (PGPR) is the biocontrol of phytopathogens. Some PGPR metabolize phytohormone abscisic acid (ABA); however, the role of this trait in plant–microbe interactions is scarcely understood. Phytopathogenic fungi produce ABA and use this property as a negative regulator of plant resistance. Therefore, interactions between ABA-producing necrotrophic phytopathogen *Botrytis* sp. BA3 with ABA-metabolizing rhizobacterium *Rhodococcus* sp. P1Y were studied in a batch culture and in gnotobiotic hydroponics with sunflower seedlings. Rhizobacterium P1Y possessed no antifungal activity against BA3 and metabolized ABA, which was synthesized by BA3 in vitro and in associations with sunflower plants infected with this fungus. Inoculation with BA3 and the application of exogenous ABA increased the root ABA concentration and inhibited root and shoot growth, suggesting the involvement of this phytohormone in the pathogenesis process. Strain P1Y eliminated negative effects of BA3 and exogenous ABA on root ABA concentration and plant growth. Both microorganisms significantly modulated the hormonal status of plants, affecting indole-3-acetic, salicylic, jasmonic and gibberellic acids, as well as cytokinins concentrations in sunflower roots and/or shoots. The hormonal effects were complex and could be due to the production of phytohormones by microorganisms, changes in ABA concentrations and multiple levels of crosstalk in hormone networks regulating plant defense. The results suggest the counteraction of rhizobacteria to ABA-producing phytopathogenic fungi through the metabolism of fungal ABA. This expands our understanding of the mechanisms related to the biocontrol of phytopathogens by PGPR.

## 1. Introduction

A variety of plant-associated bacteria including plant growth-promoting rhizobacteria (PGPR) possess a set of properties that have a negative impact on deleterious microorganisms and facilitate the biocontrol of phytopathogens. Known mechanisms of rhizobacterial effects on phytopathogens include the production of antibiotics, lytic enzymes and other antimicrobial substances, competition for nutrient sources (e.g., for root exudates and for iron due to the production of siderophores) and ecological niches in the root zone, as well as the detoxification and degradation of virulence factors [1,2]. Biocontrol mechanisms can also be directed at the plant itself, namely enhancing plant defense responses such as induced systemic resistance (ISR), systemic acquired resistance (SAR) and other effects [3,4,5]. Positive traits of plant growth-promoting rhizobacteria (PGPR), such as phytohormone production, 1-aminocyclopropane-1-carboxylate (ACC) deaminase activity, and improving mineral nutrition can stimulate plant growth and contribute to the increased disease resistance of various plant species [6,7].

Biocontrol effects of rhizobacteria were described for sunflower (*Helianthus annuus* L.). Inoculation with *Pseudomonas guariconensis* LE3 producing antibiotics, siderophores, hydrolytic enzymes, auxins and ACC deaminase stimulated growth and protected sunflower plants against charcoal rot disease caused by *Macrophomina phaseolina* [8]. Strain *Ps. aeruginosa* PF23 decreased disease symptoms of *M. phaseolina* on sunflower, whereas its mutant that is deficient in the production of exopolysaccharides and salicylic acid (SA) showed no biocontrol traits [9]. Biocontrol of sunflower against *Fusarium oxysporum* was described for PGPR strains *Pseudomonas* sp. AF-54 producing hydrolytic enzymes [10] and *Priestia koreensis* LV19 producing auxins and possessing antifungal activity in vitro [11]. Strain *Bacillus subtilis* S-16 isolated from the sunflower rhizosphere produced volatile compound 2-methyl benzothiazole (2-MBTH) and inhibited the growth of *Sclerotinia sclerotiorum*; however, the mutant producing less 2-MBTH had a decreased ability to protect the infected sunflower leaves [12]. Another strain, *Ps. putida* MS6, from the sunflower rhizosphere produced auxins, ACC deaminase, siderophores, hydrolytic enzymes and antibiotics, and showed antifungal activity in vitro against various phytopathogenic species [13].

Necrotrophic phytopathogen *Botrytis cinerea* (Ascomycota) has a wide range of plant hosts and infects sunflower, causing gray mold [14,15,16]. All the above-mentioned biocontrol mechanisms of rhizobacteria against *B. cinerea* and successful applications of such biological PGPR agents were repeatedly described for various plant species [17,18,19]. Nevertheless, information about biocontrol effects of rhizobacteria on sunflower plants infected by *B. cinerea* is limited.

Abscisic acid (ABA), being one of the major phytohormones involved in the adaptation of plants to abiotic stresses, plays an important role in plant relations with phytopathogens [20]. Various phytopathogenic fungi, including *B. cinerea*, produce ABA [21] and as a rule, fungal ABA is considered as a negative regulator of plant resistance to pathogenesis processes [22]. Accumulation of ABA in plants directly interferes with SAR and ISR by acting as an antagonist of the defense-signaling hormones SA, jasmonic acid (JA) and ethylene, which leads to increased susceptibility of plants to pathogens [23]. Increased tolerance of tomato ABA-deficient mutant *sitiens* to *B. cinerea* was associated with the inhibition of the salicylate defense pathway [24], altered composition and low permeability of the cuticle to the pathogen [25], increased activity of glutamine synthetase and gamma-aminobutyric acid metabolism [26], and the rapid induction of the biosynthesis of nitric oxide, ethylene [27] and hydrogen peroxide [28]. ABA-deficient mutants of *Arabidopsis thaliana aba3-1* or *aba1-6* showed reduced susceptibility to *Ps. syringae* [29] or *Plectosphaerela cucumerina* [30], respectively. Mutant *A. thaliana* WRKY33 with increased sensitivity to *B. cinerea* had increased ABA concentrations due to negative signaling determined by genes *NCED3* and *NCED5* involved in ABA biosynthesis [31]. Treatment with exogenous ABA decreased the resistance of *A. thaliana* to *Ps. syringae* [29], rice to various fungal pathogens [32,33,34] and tobacco to *Ralstonia solanacearum* [35]. Germination of *B. cinerea* [36] and *Magnaporthe oryzae* [34] spores was stimulated by ABA treatment. These studies demonstrate the relevance of searching for biological approaches to increasing plant resistance to phytopathogens by regulating the concentration of ABA in infected plants.

Rhizobacteria are capable of not only synthesizing phytohormones, but also metabolizing these substances, using them as a nutrient source [37]. Several bacteria capable of metabolizing ABA have now been described. The first mentioned ABA-metabolizing bacterium was *Corynebacterium* sp. isolated from ABA-supplemented soil, which exhibited vomifoliol dehydrogenase activity and converted ABA to dehydrovomifoliol [38]. However, its characteristics and interaction with plants was not further studied. Strains *Rhodococcus* sp. P1Y and *Novosphingobium* sp. P6W were isolated from the rhizosphere of rice using a selective medium containing ABA as a sole carbon source and were able to decrease root ABA concentrations in inoculated rice and tomato seedlings [39]. These rhizobacteria metabolized ABA successively into dehydrovomifoliol [40] and rhodococcal acid [41]. The ABA-metabolizing strain *R. qingshengii* (BNCC203056) increased the uptake of heavy metals by *A. thaliana* and several hyperaccumulating plant species [42,43]. The recently isolated pseudomonads *Ps. plecoglossicida* 2.4-D [44], *Ps. veronii* IBK11-1, *P. frederiksbergensis* IBTa10m and *Pseudomonas* sp. TaE2 [45] utilized ABA and decreased ABA concentrations in lettuce and wheat shoots, abolishing competitive effects in high-density plantings. Although the data about ABA-metabolizing bacteria are very limited, the available results show the potential for their use in the phytoremediation of contaminated soils and for plant growing. However, information on the interaction of such bacteria with phytopathogenic microorganisms and plant protection is still lacking.

We hypothesized that the negative effect of phytopathogenic fungi caused by ABA production could be counteracted by inoculation with ABA-metabolizing rhizobacteria. Thus, sunflower plants were treated with ABA-producing fungus *Botrytis* sp. strain BA3 and ABA-metabolizing rhizobacterium *Rhodococcus* sp. P1Y. Treatment with exogenous ABA was used to simulate the effect of fungal ABA on plants and to confirm that the beneficial effect of rhizobacteria was mediated by this phytohormone.

## 2. Results

### 2.1. Properties of Microorganisms

The sequence of the ITS region of the BA3 strain (PQ591697.1) showed varying degrees of similarity to several species within the genus *Botrytis* (Appendix A). The sequence exhibited 100% similarity and 98% query coverage with the *Botrytis eucalypti* strain CERC 7170 (KX301016.1), and 100% similarity and 90% query coverage with both the *Botrytis pelargonii* (AJ716290.1) and *Botrytis fabae* (MH855020.1) strains. Additionally, strain BA3 showed high ITS sequence similarity with other species such as *B. caroliniana*, *Botryotinia ranunculi*, *B. californica*, *B. deweyae*, *B. sinoallii*, *Botryotinia polyblastis*, and *B. porri*, with similarities ranging from 99.02% to 99.80% and query coverage from 90% to 100% (Appendix A). Based on these results, the studied fungal strain BA3 was assigned to the genus *Botrytis* without a specific species definition.

Experiments in vitro showed the ability of *Rhodococcus* sp. P1Y to produce IAA and ILA, whereas *Botrytis* sp. BA3 produced ABA, indole-3-acetic acid (IAA), indole-3-carboxylic acid (ICA), indole-3-lactic acid (ILA), gibberellic acid (GA3) and SA (Table 1). Relatively small amounts of dihydrozeatin (DHZ) and *trans*-zeatin (tZ) were detected in supernatants of both microorganisms (Table 1). No growth of both microorganisms was detected on the medium containing IAA or SA as a carbon source and ACC as a nitrogen source.

### 2.2. Combined Batch Culture

The number of *Rhodococcus* sp. P1Y at the 14-th day of combined cultivation with *Botrytis* sp. BA3 was lower than that in pure culture, suggesting the growth inhibition of bacteria caused by the fungus (Figure 1A). Growth of *Botrytis* sp. BA3 was not affected by the presence of *Rhodococcus* sp. P1Y (Figure 1B). Strain *Botrytis* sp. BA3 produced ABA with a maximal concentration of 451 ± 33 nM at the 14-th day, whereas *Rhodococcus* sp. P1Y actively consumed fungal ABA (Figure 1C).

### 2.3. Selection of Exogenous ABA Concentrations for Sunflower

Treatment of sunflower seedlings with exogenous ABA inhibited root (Figure 2A) and shoot (Figure 2B) growth in a dose-effect manner. The decrease in both root and shoot biomass was observed at 400 nM ABA. Since approximately such an ABA concentration was also detected in batch culture of *Botrytis* sp. BA3 (Figure 1C), this concentration was chosen for further experiments.

### 2.4. Effect of Microorganisms on Sunflower Growth

Inoculation of sunflower seedlings with *Botrytis* sp. BA3 decreased the biomass of roots (Figure 3A) and shoots (Figure 3B). Treatment with ABA also inhibited plant growth with a maximal negative effect on shoots in the presence of *Botrytis* sp. BA3 (Figure 3B). Inoculation with single *Rhodococcus* sp. P1Y did not affect plant growth, but this strain eliminated the negative effect of *Botrytis* sp. BA3 and exogenous ABA. When the plants were both inoculated with *Botrytis* sp. BA3 and treated with ABA, the growth-promoting effect of *Rhodococcus* sp. P1Y was 72% and 83% on roots and shoots, respectively (Figure 3). Examples of uninoculated and inoculated plants are shown in Figure 4. Inoculation with *Botrytis* sp. BA3 impeded the growth of the main root and caused the maceration of its tissues (Figure 5A). Dark-gray necroses (Figure 5B,C) were formed on the cotyledons of few plants, in which the fungal mycelium was visible (Figure 5D). The dying tissues at the initial stages of disease had a greenish-brown color due to the residual amount of chlorophyll (Figure 5E), but then turned black and contained the fungal mycelium (Figure 5F) in such cotyledons. The control plants and those inoculated with *Rhodococcus* sp. P1Y or treated with ABA had no such disease symptoms.

### 2.5. Presence of Microorganisms in Hydroponic Culture

In the end of experiments, the number of *Rhodococcus* sp. P1Y in the nutrient solution was approximately similar in all treatments, although inoculation with *Botrytis* sp. BA3 decreased the number of bacteria by 41% in the presence of exogenous ABA (Table 2). No significant differences were observed in the amount of fungal DNA on roots (Table 2). Contamination with other microorganisms was not detected in the nutrient solution nor on roots.

### 2.6. Effect of Microorganisms on ABA Concentrations

The concentration of ABA in the nutrient solution supplemented with *Botrytis* sp. BA3 or with exogenous ABA in the end of experiments was about 40 nM, whereas it was twice that amount (103 nM) when both components were present (Figure 6). In these treatments, strain *Rhodococcus* sp. P1Y decreased the ABA concentration in the nutrient solution by many times. The ABA concentration in the nutrient solution negatively correlated with root (r = −0.92, *p* = 0.001, n = 8) and shoot (r = −0.93, *p* = 0.001, n = 8) biomass. Other phytohormones (IAA, SA, JA, GA3, DHZ, tZ and tZR) were not detected in the nutrient solution of un-inoculated and inoculated variants.

Root ABA concentration increased after inoculation with *Botrytis* sp. BA3 and/or treatment with exogenous ABA (Figure 7A). Such an effect was completely eliminated in the presence of *Rhodococcus* sp. P1Y, restoring the ABA concentration in roots to the level of un-inoculated plants. The shoot ABA concentration was not affected by single inoculations with the studied microorganisms or by the treatment with ABA (Figure 7B). At the same time, increased ABA concentration in shoots was detected when *Rhodococcus* sp. P1Y was applied with ABA treatment and/or with *Botrytis* sp. BA3 inoculation. ABA concentration in roots positively correlated with ABA concentration in nutrient solution (r = +0.92, *p* = 0.001, n = 8), but no correlation was found between shoot and solution ABA concentrations.

### 2.7. Effect of Microorganisms on Other Hormones in Sunflower

Treatment with only exogenous ABA had little effect on concentrations of other phytohormones measured in sunflower plants with the exception of lowering SA and tZ in roots, as well as GA3 and DHZ in shoots (Table 3). Strain *Rhodococcus* sp. P1Y increased the IAA concentration in roots of ABA-untreated plants. Strain *Botrytis* sp. BA3 alone and a combined inoculation with both strains increased concentrations of JA, GA3, DHZ and tZR, but decreased tZ concentration in roots of ABA-untreated plants (Table 3). Increased SA and tZ concentrations in roots inoculated with *Rhodococcus* sp. P1Y, as well as SA, JA and tZR concentrations in roots inoculated with *Botrytis* sp. BA3, were observed in ABA-treated plants. Mixed inoculation decreased IAA and DHZ concentrations, but increased the SA concentration in these roots.

Shoots of ABA-untreated plants showed decreased GA3 and DHZ concentrations after inoculation with *Rhodococcus* sp. P1Y, increased JA and GA3 concentrations after inoculation with *Botrytis* sp. BA3 and decreased DHZ concentration after inoculation with any strain (Table 3). Inoculation with *Rhodococcus* sp. P1Y increased JA concentration, whereas combined inoculation increased IAA, SA, JA, GA3, tZ and tZR concentrations in shoots of ABA-treated plants.

## 3. Discussion

### 3.1. Interactions Between Microorganisms

The genus *Botrytis* has wide species diversity with many species being quite similar taxonomically [46]. Strain BA3 showed close similarity to several *Botrytis* species including *B. cinerea*, therefore it was identified on a genus level as *Botrytis* sp. In bath culture, *Botrytis* sp. BA3 produced about 225 ÷ 400 nM of ABA (59 ÷ 106 µg ABA L^−1^) (Table 1; Figure 1C), that was comparable with the data for other ABA-producing fungal species available in the literature, specifically *B. cinerea* [21,47]. Both microorganisms produced auxins, GA3, DHZ and tZ, suggesting possibility for involvement of these traits in modulating plant growth. Production of auxins by *Rhodococcus* sp. P1Y was previously reported [39] and here it was used as a positive control. Both microorganisms did not utilize IAA and SA, whereas *Rhodococcus* sp. P1Y did not utilize ABA. Both microorganisms did not grow in the presence of ACC as a nitrogen source, suggesting the absence of ACC deaminase activity. Therefore, these properties can be avoided from discussion related to plant–microbial interactions.

Combined batch cultivation showed that *Rhodococcus* sp. P1Y did not affect the growth of *Botrytis* sp. BA3 (Figure 1B), suggesting an absence of antagonistic and antifungal activity of bacteria against fungus. A lower number of bacteria in the presence of *Botrytis* sp. BA3 could probably be due to competition for nutrients, since the effect was moderate and temporary. The results proved the ability of *Rhodococcus* sp. P1Y to utilize ABA, which was synthesized by *Botrytis* sp. BA3 in combined culture. Evaluation of such interactions between rhizobacteria and phytopathogenic fungi in vitro was important for understanding the mechanisms of the biocontrol effect of rhizobacteria in experiments with sunflowers.

### 3.2. Effects of Exogenous ABA on Plants

Although the growth-stimulating role of ABA for plants was discussed [48,49], the negative effects of high exogenous ABA concentrations on plant physiology were also reported [49,50,51,52]. Particularly, treatment with ABA inhibited the root growth of *A. thaliana* [53,54,55] and several agricultural crops such as lentil [56], soybean [57], rice [58], peanut [59] and tomato [60]. This report expands such an effect for sunflowers. The mechanisms of the growth inhibitory activity of endogenous ABA were also demonstrated in ABA-deficient and insensitive mutants [49]. The expected result was that exogenous ABA increased the concentration of this phytohormone in sunflower roots (Figure 7A). The negative effect of exogenous ABA on sunflower growth was probably associated with a disruption of the hormonal status of plants. This was manifested in decreased SA and tZ concentrations in roots, as well as the GA3 and DHZ concentration in shoots of ABA-treated plants (Table 3). Negative physiological changes related to the imitation of abiotic stress [49] and decreased root hair length and density [58,61,62,63] could contribute to the disturbance of metabolism and the restriction of water and nutrient uptake by plants. The mechanisms of action of exogenous ABA that are significant for sunflowers remain to be elucidated. Deciphering these mechanisms will help to find approaches to protect plants against ABA-producing phytopathogens.

### 3.3. Effect of Microorganisms on Plants

The decrease in the biomass of roots and shoots was achieved in the applied exogenous ABA concentrations (Figure 2) similar to those produced in vitro by *Botrytis* sp. BA3 (Figure 1C), suggesting that fungal ABA could be involved in plant growth inhibition. Similar ABA concentrations were detected in the nutrient solution treated with ABA and inoculated with *Botrytis* sp. BA3 at the end of experiments with sunflowers (Figure 6). However, *Botrytis* sp. BA3 inhibited plant growth to a greater extent compared to exogenous ABA (Figure 3). Damage to root and shoot tissues caused by *Botrytis* sp. BA3 also demonstrated the development of disease symptoms on the inoculated plants (Figure 5). It is known that phytopathogenic fungi of the genus *Botrytis* produce toxins (botrydial, dihydrobotrydial, botcinolide and botcinic acid), plant cell-wall-degrading enzymes and reactive oxygen species [64,65]. It is most likely that *Botrytis* sp. BA3 has such properties; however, more detailed study is needed in the future to describe the mechanisms of pathogenesis that are inherent to this strain. In the present study, only the root ABA concentration negatively correlated with the effect of treatments on both root and shoot biomass (Appendix A), demonstrating the major role of this phytohormone in plant growth regulation. The revealed correlations between root biomass and ABA concentrations in nutrient solution or in sunflower roots also showed that the production of ABA by *Botrytis* sp. BA3 could contribute to the growth inhibition of sunflower plants.

Inoculation with *Rhodococcus* sp. P1Y decreased the ABA concentration in both the nutrient solution (Figure 6) and in sunflower roots (Figure 7A) most likely due to the metabolism of this compound since this strain was previously described as an ABA-utilizing rhizobacterium [40,41]. This effect combined with the removal of the inhibitory action of ABA and *Botrytis* sp. BA3 on sunflower root and shoot growth (Figure 3). The presence of *Rhodococcus* sp. P1Y did not reduce root colonization by *Botrytis* sp. BA3 (Table 2), confirming the absence of bacterial antagonism and antifungal activity against fungus observed in vitro (Figure 1). In addition, disease symptoms on sunflower shoots (Figure 5) were observed on plants inoculated with single *Botrytis* sp. BA3 only. Taken together, these results suggest that the biocontrol activity of *Rhodococcus* sp. P1Y was due to the elimination of the negative impact of fungal ABA on plants, leading to the attenuating virulence of the fungus. Several mechanisms of the biocontrol of phytopathogenic fungi by rhizobacteria are currently known. They are based on the production of antimicrobial substances (antibiotics, toxic compounds, lytic enzymes), competition for nutrients and ecological niches, the detoxification and degradation of virulence factors, and the induction of ISR and SAR [1,2,3,4,5]. Successful applications of such rhizobacteria were described for sunflowers infected by various phytopathogenic fungi such as *M. phaseolina* [8,9], *F. oxysporum* [10,11,13], *F. moniliforme*, *Rhizoctonia solani*, *Colletotrichum gloeosporioides*, *C. falcatum* [13] and *S. sclerotiorum* [12]. Necrotrophic phytopathogen *B. cinerea*, which is the closest relative of *Botrytis* sp. BA3, was repeatedly and efficiently suppressed in some plant species by inoculation with rhizobacteria possessing the properties mentioned above [17,18,19]. However, limited information is available about the biocontrol of sunflower against *B. cinerea*. For the first time, this study describes the ability of rhizobacteria to counteract negative effects of the ABA-producing necrotrophic phytopathogen belonging to *Botrytis* on the host plant by means of ABA metabolism.

### 3.4. Effect of Exogenous ABA and Microorganisms on Hormonal Status of Plants

Dissimilarity between concentrations of IAA, SA, GA3, DHZ and tZ in roots inoculated with *Rhodococcus* sp. P1Y and infected by *Botrytis* sp. BA3 or treated with ABA were found (Table 3), highlighting differences in plant response to these microorganisms. Particularly, the similarity between the effects of exogenous ABA and *Botrytis* sp. BA3 was manifested in the decrease in root SA and tZ (Table 3). This result is in line with information about antagonism between ABA and SA in plants infected with phytopathogens, leading to the interference of defense signaling [23,24]. Opposite responses to ABA or *Botrytis* sp. BA3 compared with *Rhodococcus* sp. P1Y were also observed. However, the complexity and ambiguity in such interpretation was evident from the heat map which showed variants ABA, *Botrytis* sp. BA3 and *Rhodococcus* sp. P1Y grouping in one cluster IV in roots by the plant hormonal profile (Figure 8A) and cluster VIII in shoots (Figure 8B), respectively. The most striking similarity of treatments was observed for groups of variants ABA and *Rhodococcus* sp. P1Y in the presence and absence of *Botrytis* sp. BA3 (Figure 8A cluster III), as well as variant *Botrytis* sp. BA3 in the presence and absence of ABA or *Rhodococcus* sp. P1Y combined in cluster IV (Figure 8A). Hormonal profiles were divided into two clusters depending on the treatments, one of which combined JA, tZR and ABA (cluster II), and the other combined the remaining hormones (cluster I). The increase in the concentration of ABA was accompanied by an increase in the concentration of JA, but this did not depend on the treatment options for the plants. Perhaps this effect was due to the synergistic crosstalk of these hormones, which has been discussed in many studies (for example, [66,67,68]). The complexity of hormonal relations in sunflower roots was also evidenced by the absence of correlations between the concentrations of phytohormones, with the exception of positive correlations of tZR with ABA and JA, as well as DHZ with GA3 (Appendix A). Thus, analysis of phytohormone concentrations in sunflower roots revealed very complex effects of microorganisms and exogenous ABA, probably due to the interaction of hormones.

Single *Botrytis* sp. BA3 decreased SA concentration in roots, whereas both strains increased the concentration of this phytohormone when plants were treated with ABA. The concentration of GA3 increased in roots inoculated with *Botrytis* sp. BA3 and the shoot DHZ of all inoculated plants decreased only in the absence of exogenous ABA (Table 3). The observed effect was probably due to the production of GA3 by the fungus, since it was a very active producer of this phytohormone in vitro (Table 1). In addition, *Rhodococcus* sp. P1Y increased the IAA concentration in the ABA-untreated roots only and opposite effects of combined inoculation on the root IAA concentration were found (Table 3). Comparison of the concentrations of root phytohormones in *Rhodococcus* sp. P1Y variants treated with ABA or in the presence of *Botrytis* sp. BA3 revealed significant differences for IAA, GA3, DNZ, tZ and tZR (Table 3). Moreover, these variants grouped in different clusters, III and IV, by phytohormone profiles (Figure 8A). This indicated significant differences in the effects of ABA and *Botrytis* sp. BA3 on the hormonal changes induced by *Rhodococcus* sp. P1Y, despite the similarity of the effects of these agents in the absence of rhizobacteria (Table 3, Figure 8A). Thus, exogenous ABA significantly changed the effect of microorganisms on the concentration of phytohormones in sunflower roots.

Both microorganisms produced IAA also, suggesting the possibility for the involvement of this trait in modulating plant growth since this trait is well known as a plant growth-promoting factor [69,70]. However no stimulating effect of microorganisms was observed in experiments with sunflowers (Figure 3) and no correlations were found between IAA concentrations and plant biomass (Appendix A). It is possible that the auxin-producing activity of *Rhodococcus* sp. P1Y increased IAA concentration in roots (Table 3), but was too low to stimulate sunflower growth. The possible effect of *Botrytis* sp. BA3 related to IAA production was neutralized by its pathogenic properties.

Cytokinin concentrations in the studied plants were comparable with those reported previously for sunflower young plants [71]. Changes in plant cytokinins caused by inoculations were probably not due to production by microorganisms because (1) only trace and similar amounts of DHZ were found in batch cultures of both strains (Table 1), whereas increased an DHZ concentration was observed in roots inoculated with *Botrytis* sp. BA3 only; (2) *Botrytis* sp. BA3 produced tZ, but decreased its concentration in roots; (3) *Botrytis* sp. BA3 did not produce tZR (Table 1); (4) *Rhodococcus* sp. P1Y produced DHZ and tZ (Table 1), but did not affect their root concentrations. Moreover, the root tZR concentration negatively correlated with the effect of treatments on shoot biomass, whereas no correlations were found between other cytokinin concentrations and plant biomass (Appendix A). Although microbial cytokinins could play an important role in the growth promotion of the inoculated plants [70], our results suggest that this trait was not involved in the sunflower growth response to the applied inoculations.

In general, the intensity of the effect of exogenous ABA and microorganisms on the concentration of phytohormones in shoots was less than that in roots (Table 3, Figure 8B). This was expected, since the roots but not shoots were treated. Inoculation with *Rhodococcus* sp. P1Y increased ABA concentration in shoots only in the presence of ABA-producing *Botrytis* sp. BA3 or exogenous ABA (Figure 7B). This could likely be due to the bacterial effect on the transport of exogenous ABA from root to shoot, since previously this strain did not increase endogenous ABA in shoots of rice and tomato seedlings [39]. However, the mechanism of the observed effect of *Rhodococcus* sp. P1Y needs further, more detailed investigation. Increased shoot ABA concentrations were previously reported after inoculation with PGPR of various plants, such as tomato with *B. pumilus* ACCC19290 [72], *Ps. fluorescens* G20-18 [73], *Ps. aeruginosa* HG28-5 [74] and *Leclercia adecarboxylata* MO1 [75]. But the effect of these PGPR on ABA concentrations in sunflowers received little attention and ABA production or metabolism by these strains was not studied. It should be noted that the increase in the concentration of GA3 in shoots (as well as in roots) infected with *Botrytis* sp. BA3 (Table 3) could also be associated with the active production of this phytohormone by the fungus (Table 1). The similarity of the shoot and root response of plants to inoculation with *Botrytis* sp. BA3 was manifested in an increase in JA concentrations (Table 3). This indicates the activation of JA-dependent responses and the systemic protective effect caused by *Botrytis* sp. BA3 that is typical for plant interactions with necrotrophic phytopathogens [76]. Interestingly, only the combined treatment with ABA and both microorganisms lead to the increased shoot concentration of all phytohormones, except DHZ (Table 3). It is likely that the combination of stimulants modulated hormonal status to a greater extent and worked as very complex signals affecting a network of different metabolic pathways [67,68,70,77].

Considering these results together, explaining the modulations of plant hormonal status in such a multi-component system involving exogenous ABA and microbial metabolizers and/or producers of various hormones is a complex task. We have limited ourselves to a brief description and interpretation of the observed hormonal changes induced by exogenous ABA and the studied microorganisms, emphasizing those related to ABA. A more thorough understanding of these results and testing of the hypotheses requires a detailed analysis of the rich literature in this area and additional research.

## 4. Materials and Methods

### 4.1. Plants and Microorganisms

Seeds of sunflower (*Helianthus annuus* L.) cultivar Surus were obtained from the Federal Scientific Center All-Russia Research Institute of Oil Cultures named after V.S. Pustovoit. Bacterial strain *Rhodococcus* sp. P1Y [39] capable of metabolizing ABA and fungal strain *Botrytis* sp. BA3 (initially isolated from wheat seeds originated from the Republic of Bashkortostan in 2017) capable of producing ABA were obtained from the Russian Collection of Agricultural Microorganisms (RCAM, St.-Petersburg, Russian Federation, http://www.arriam.ru/kollekciya-kul-tur1/; assessed on 3 April 2024) and maintained in the course of experimental work on potato dextrose broth (PDB) or agar (PDA) medium (HiMedia Laboratories Pvt. Ltd., Mumbai, India). The ABA-producing trait of *Botrytis* sp. BA3 was discovered (data not published) during a screening process of other collection fungal strains using a bath culture as described previously [21]. This strain was identified by the determination of the nuclear ribosomal internal transcribed (ITS) region sequence [78]. The sequence was compared to similar sequences in the NCBI database using BLAST (https://blast.ncbi.nlm.nih.gov/Blast.cgi accessed on 8 August 2024) analysis to identify the closest relatives and then it was submitted to the NCBI GenBank database (sequence accession number PQ591697.1).

### 4.2. Batch Cultures

To study in vitro interactions between *Rhodococcus* sp. P1Y and *Botrytis* sp. BA3, the microorganisms were grown as pure and mixed cultures in flasks with 50 mL of the original nutrient medium OCD containing (mg L^−1^) glucose—10,000; glutamic acid—200; asparagine—200; aspartic acid—200; serine—200; thiamine—1; MgSO_4_—200; KCl—500; CaCl_2_—100; KH_2_PO_4_—800; FeCl_3_—50; H_3_BO_3_—0.12; MnSO_4_—0.15; ZnSO_4_—0.16; NaCl—0.06; Na_2_MoO_4_—0.18; CoCl_2_—0.13; CuCl_2_—0.13; NiCl_2_—0.13; and with a pH = 6.0 [21]. Bacteria were added in a final concentration of 5 × 10^5^ cells mL^−1^. To obtain the inoculum of *Botrytis* sp. BA3, the fungus was grown in Petri dish with PDA medium for 7 days. Then pieces of mycelium with agar of approximately 4 mm^2^ were cut out and placed into flasks. Nine flasks were prepared for each treatment. Cultivation of the studied microorganisms was carried out for 21 days in the dark at 25 °C without shaking. Three flasks were taken for analysis every week of cultivation. The number of bacteria in the culture fluid was determined by the standard dilution method using PDA medium. The fungal biomass was dried and weighed. The culture fluids were centrifuged for 15 min at 9000× *g* and 4 °C, acidified with 1 N HCl to a pH = 3.0, extracted with equal volumes of ethyl acetate and stored at −20 °C for further determination of ABA concentrations. The experiment was conducted two times.

To study the production of phytohormones, the microorganisms were cultivated for 7 days in PDB medium supplemented with 100 mg L^−1^ of tryptophan as described above. The culture fluids were centrifuged, acidified, extracted and stored as described above. The ability to utilize phytohormones was studied via the cultivation of the microorganisms for 7 days on mineral OCD medium containing ABA, IAA, or SA as a sole source of carbon and 200 mg L^−1^ of NH_4_NO_3_ as a nitrogen source. No ammonium nitrate was added when microorganisms were cultivated in the presence of 1-aminocyclopropane-1-carboxylic acid (ACC). The complete OCD medium was used as a positive control, whereas the OCD medium containing no carbon or nitrogen source was used as a negative control. Bacterial growth was determined by measuring the optical density of suspensions at 540 nm against corresponding negative controls using spectrophotometer SmartSpec Plus (Bio-RAD, Hercules, CA, USA). Microorganisms were cultivated in two replicates for each experiment.

### 4.3. Hydroponic Culture with Sunflower

Seeds of sunflower were surface-sterilized by successive treatment with 70% ethanol for 1 min and 3% sodium hypochlorite for 6 min, rinsed with sterile tap water and germinated in Petri dishes for two days at 27 °C. Then five seedlings were transferred to each polypropylene pot (OS140BOX, Duchefa, The Netherlands) containing 250 mL of sterile nutrient solution (µM): KNO_3_, 1200; Ca(NO_3_)_2_, 60; MgSO_4_, 250; KCl, 250; CaCl_2_, 60; Fe-tartrate, 12; H_3_BO_3_, 2; MnSO_4_, 1; ZnSO_4_, 3; NaCl, 6; Na_2_MoO_4_, 0.06; CoCl_2_, 0,06; CuCl_2_, 0.06; NiCl_2_, 0,06; pH = 6.5. The seedlings were placed on stainless steel mesh so that only the roots were immersed in the solution. The nutrient solution was supplemented or not with different concentrations of ABA ranging from 100 to 50,000 nM and with bacterium *Rhodococcus* sp. P1Y in a final concentration of 10^5^ cells mL^−1^ and/or with fungus *Botrytis* sp. BA3 in the amount of 10^6^ spores mL^−1^. Non-supplemented solution was used as a control treatment.

Plants were cultivated for 10 days in a growth chamber (ADAPTIS-A1000, Conviron, Isleham, UK) with 200 µmol of quanta m^−2^ s^−1^, a 12 h photoperiod and minima–maxima temperatures of 18–23 °C. An example of a pot with plants is shown in the online resource (Appendix A). Then, the fresh root and shoot biomass of individual plants was determined, the plants were immediately frozen in liquid nitrogen and stored at −80 °C for further determination of phytohormones. A part of the roots was dried at room temperature and used to estimate the presence of *Botrytis* sp. BA3. The nutrient solution was centrifuged for 15 min at 9000× *g* and 4 °C and also used for further determination of phytohormones. Images of roots and leaves (Figure 5) were taken using a stereo microscope Stemi 508 (Carl Zeiss, Oberkochen, Germany). The experiment with 100÷400 nM ABA concentrations was conducted two times and the experiment with 2000÷5000 nM ABA concentrations was conducted one time with two pots per each treatment. Experiments with microorganisms were repeated three times with three pots per each treatment.

### 4.4. Presence of Microorganisms in Hydroponic Culture

The nutrient solution was checked for contamination at the 5-th day after inoculation via the incubation of aliquots of the nutrient solution onto PDA medium. at the end of experiments, the number of *Rhodococcus* sp. P1Y in the nutrient solution was determined by the serial dilution method using PDA medium as described previously [79]. Estimation of *Botrytis* sp. BA3 presence on roots was performed using the method described previously [80]. Briefly, the dry roots were ground and DNA was extracted by a buffer based on cetyltrimethylammonium bromide (CTAB) and homogenized using the Precellys 24 homogenizer (Bertin Technologies, Montigny-le-Bretonneux, France) at 600 rpm twice for 20 s. The samples were incubated for 10 min at 42 °C and 10 min at 65 °C for protein denaturation. For the final release from proteins, the samples were treated twice with a mixture of chloroform and isoamyl alcohol (24:1). Then DNA was precipitated by treatment with isopropanol, washed with 70% ethanol, dissolved in water and stored at −20°C. Determination of the amount of DNA of *Botrytis* sp. BA3 in the samples was carried out by RT PCR using a qPCRmix-HS SYBR reagent mixture (Evrogen, Moscow, Russia) and the specific primers 5′GCGGAGGAAGAATCATTACAG3′ and 5′GAGTTTTGGTATTCTCGGCG3′. The cycling conditions included 40 cycles of 25 s at 94 C, 25 s at 55 °C and 25 s at 72 °C. A calibration curve was constructed using pure DNA of *Botrytis* sp. BA3. Four biological and five analytical replicates of DNA determination were applied for each sample.

### 4.5. Determination of Phytohormones

Determination of ABA, auxins, SA, JA and GA3 concentrations in batch cultures was carried out by ultra-performance liquid chromatography using a Waters ACQUITY UPLC H-class system (Waters, Milford, MA, USA) on a Waters ACQUITY UPLC BEH Shield RP18 (Waters, Milford, MA, USA) column as previously described [21].

For the determination of endogenous plant hormones, the biomass of roots and shoots was homogenized in methanol using a RayKol AH-50 automatic homogenizer (RayKol Group (XiaMen) Co., Ltd., Xiamen, China). The resulting homogenate was centrifuged at 11,000 rpm and 4 °C, and the supernatant was separated and evaporated to dryness at 35 °C in a nitrogen stream using a RayKol Auto EVA-30 automatic evaporator (RayKol Group (XiaMen) Co., Ltd., Xiamen, China). The resulting dry residues were dissolved in 20% acetonitrile. The samples were pre-centrifuged at 14,000 rpm and 4 °C and filtered through a nylon membrane filters. The resulting solution was analyzed by HPLC-MS using an IT-TOF mass spectrometer (Shimadzu Corporation, Tokyo, Japan) equipped with an ESI interface. A Waters BEH Shield RP18 column (50 × 2.1 mm, 1.7 μm) (Waters, Milford, MA, USA) was used. Chromatography was carried out at temperatures +35 °C and a flow rate of 0.3 mL min^−1^. The injection volume was 5 µL.

ABA, SA and JA were analyzed in negative electrospray ionization mode. The mobile phases were 0.1% formic acid (A) and acetonitrile with 0.1% formic acid (B). The gradient was as follows: 0.0–1.0 min, 0% B, 1.0–11.0 min, 0% → 70% B; 11.0–13.0 min, 70 → 100% B; 13–15 min, 100% B and 15–15.1 min, 100% → 0% B. The IT-TOF/MS analysis was carried out in the selected mass ranges, which were (*m*/*z*) as follows: 136.5000–138.5000 Da (for SA determination); 208.5000–210.5000 Da (for JA determination); 262.5000–264.5000 Da (for ABA determination) with a scan rate 2 spectra s^−1^. The operating parameters of the electrospray ionization sources were as follows: nebulizing gas (N_2_) flow rate, 1.5 L min^−1^; drying gas pressure, 100 kPa; CDL temperature, 200 °C; heat block temperature, 200 °C. Probe voltage was −3.5 kV in the range of 6.0–7.6 min (for SA); −2.5 kV in the range of 7.6–8.6 min (for ABA); and −1.5 kV in the range of 8.6–10.0 min (for JA). Ion accumulation time was 500 ms; detector voltage—1.6 kV. Retention times of phytohormones were JA—8.9 ± 0.3 min; ABA—7.9 ± 0.3 min; SA—6.8 ± 0.3 min. Calibration was performed in the concentration range of SA—50–5000 ng mL^−1^; ABA—25–2500 ng mL^−1^; JA—100–10,000 ng mL^−1^.

Concentrations of GA3, IAA, DHZ, tZ and *trans*-zeatin-ribozide (tZR) were analyzed in positive electrospray ionization mode. The mobile phases were 0.1% formic acid (A) and acetonitrile with 0.25% formic acid (B). The gradient was as follows: 0.0–1.0 min, 0% B, 1.0–9.0 min, 0% → 75% B; 9.0–10.0 min, 75 → 100% B; 10–11 min, 100% B and 11–11.1 min, 100% → 0% B. The IT-TOF/MS analysis was carried out in the selected mass ranges, which were (*m*/*z*) as follows: 219.9000–221.4000 Da (for *t*-zeatin determination); 352.0000–353.5000 Da (for tZR); 221.9000–223.4000 Da (for DHZ determination); 285.0000–286.5000 Da (for GA3 determination); 175.9000–176.4000 Da (for IAA determination) with a scan rate 2 spectra sec^−1^. The operating parameters of the electrospray ionization sources were as follows: nebulizing gas (N_2_) flow rate, 1.5 L min^−1^; drying gas pressure—100 kPa; CDL temperature—200 °C; heat block temperature—200 °C. Probe voltage was +1.5 kV. Ion accumulation time was 500 ms; detector voltage—1.6 kV. Retention times of phytohormones were as follows: IAA—6.0 ± 0.3 min; GA3—5.8 ± 0.3 min; tZ—4.5 ± 0.2 min; tZR—4.7 ± 0.2 min; DHZ—4.4 ± 0.2 min. Calibration was performed in the following concentration range: IAA—100–10,000 ng mL^−1^; GA3—10–1000 ng mL^−1^; tZ—10–1000 ng mL^−1^; tZR—10–1000 ng mL^−1^; DHZ—10–1000 ng mL^−1^. All the acquisitions and analyses of data were controlled by LabSolutions LCMSSolution software (release 3.80, Shimadzu Corporation, Tokyo, Japan). A standard solution of sodium trifluoroacetic acid (TFA) was used to calibrate the TOF–MS to increase mass accuracy.

### 4.6. Statistical Analysis

Statistical analysis of the data was performed using the software STATISTICA version 10 (TIBCO Software Inc., Palo Alto, CA, USA). MANOVA analysis with Fisher’s LSD test and Student’s *t* test were used to evaluate differences between means. Data visualization was performed using a heatmap approach implemented in R 4.4.0 with the ggplot2 (v. 3.5.2) and patchwork (v. 1.3.0) packages [81,82,83].

## 5. Conclusions

Rhizobacterium *Rhodococcus* sp. P1Y metabolized ABA, which was synthesized by the necrotrophic phytopathogen *Botrytis* sp. BA3 in vitro and in associations with sunflower plants infected with this fungus. In both types of experiments, no antifungal activity of *Rhodococcus* sp. P1Y was detected against *Botrytis* sp. BA3; however, the fungus inhibited the proliferation of rhizobacteria in the root zone, probably due to competition for substrates. Inoculation of sunflower with *Botrytis* sp. BA3 along with the application of exogenous ABA increased root ABA concentration and inhibited root and shoot growth, suggesting the involvement of this phytohormone in the pathogenesis process. Strain *Rhodococcus* sp. P1Y controlled root ABA concentration and eliminated growth inhibition and disease symptoms caused by *Botrytis* sp. BA3 on sunflower seedlings. Both microorganisms significantly modulated concentrations of other phytohormones, such as IAA, SA, JA, GA3, DHZ and tZR in sunflower roots and/or shoots. The observed hormonal effects could be due to several factors, namely the production of phytohormones by microorganisms, induced changes in ABA concentrations and multiple levels of crosstalk in hormone networks regulating plant defense. Thus, the ability of rhizobacteria to metabolize ABA is involved into biocontrol of the ABA-producing phytopathogen on sunflower seedlings and the application of ABA-metabolizing rhizobacteria can be useful in reducing plant diseases like gray mold.

## Figures and Tables

**Figure 1 plants-14-02442-f001:**
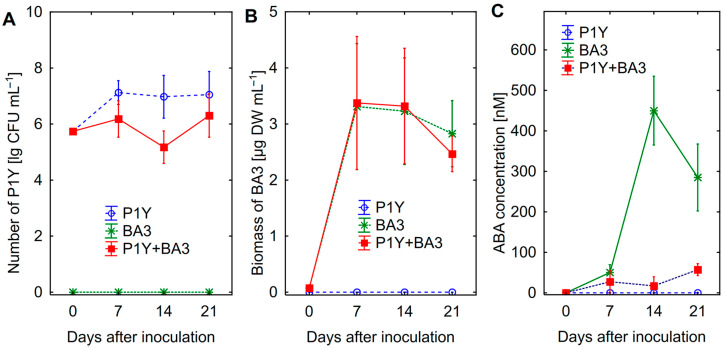
Number of *Rhodococcus* sp. P1Y (**A**), biomass of *Botrytis* sp. BA3 (**B**) and abscisic acid (ABA) concentration in the nutrient solution (**C**) in batch culture with pure cultures and a mixture of microorganisms. Vertical bars show confidence intervals (*p* = 0.05, n = 6).

**Figure 2 plants-14-02442-f002:**
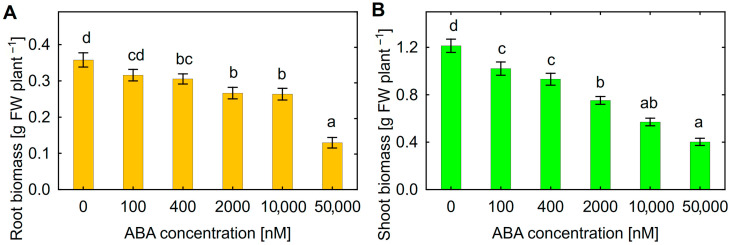
Effect of exogenous abscisic acid (ABA) on biomass of sunflower roots (**A**) and shoots (**B**). Mean data of two independent experiments. Different letters indicate significant differences between treatments (Fisher’s LSD test, *p* < 0.05, n = 20 for treatments from 0 to 400 nM ABA and n = 10 for treatments from 2000 to 50,000 nM ABA).

**Figure 3 plants-14-02442-f003:**
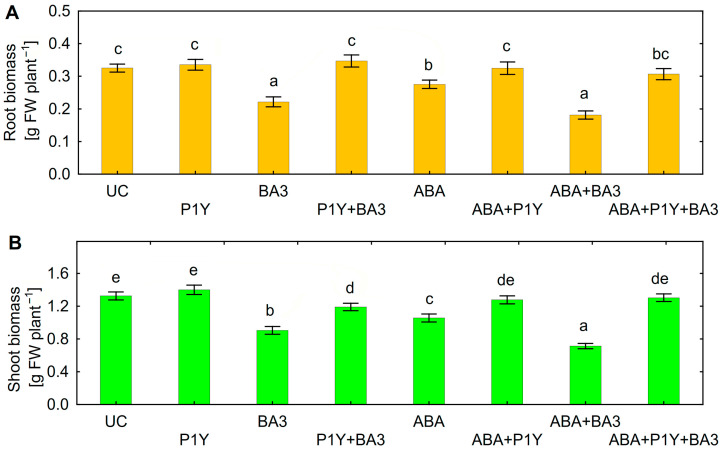
Biomass of sunflower roots (**A**) and shoots (**B**) inoculated with *Rhodococcus* sp. P1Y (P1Y) and *Botrytis* sp. BA3 (BA3) and treated with 400 nM of abscisic acid (ABA). Mean data of three independent experiments. UC stands for un-inoculated control. Different letters indicate significant differences between treatments (Fisher’s LSD test, *p* < 0.05, n = 45).

**Figure 4 plants-14-02442-f004:**
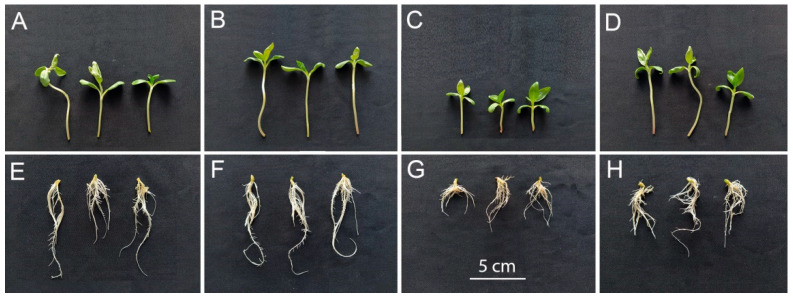
Sunflower plants uninoculated (**A**,**E**) or inoculated with *Rhodococcus* sp. P1Y (**B**,**F**), *Botrytis* sp. BA3 (**C**,**G**) and a mixture of *Rhodococcus* sp. P1Y and *Botrytis* sp. BA3 (**D**,**H**) at the end of the experiment. The scale on G is the same for all parts of the figure.

**Figure 5 plants-14-02442-f005:**
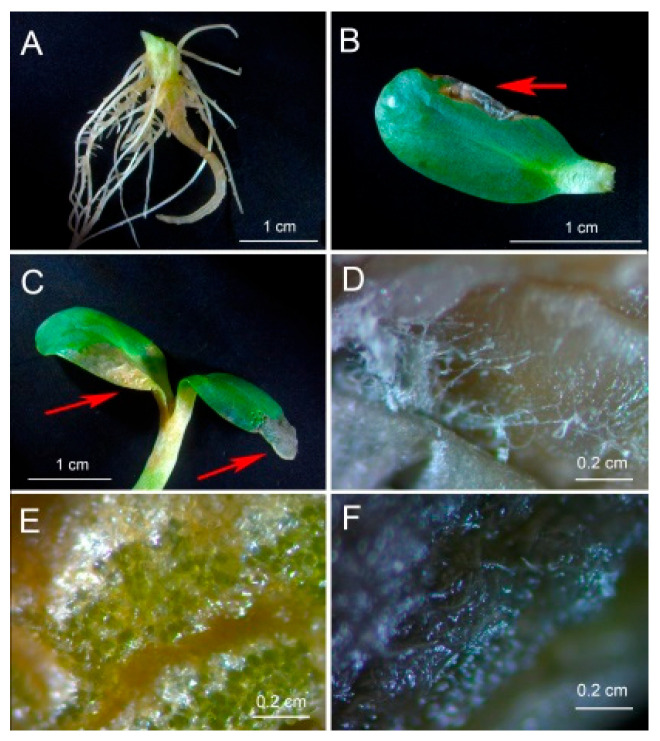
Tissue damage of sunflower roots (**A**) and shoots (**B**–**F**) inoculated with *Botrytis* sp. BA3. (**A**) A root system with inhibited growth and maceration of the main root. (**B**,**C**) Damaged shoots. (**D**) The place of mycelium formation, indicated by the red arrow in (**B**). (**E**) Greenish-brown damaged tissue with the residual amount of chlorophyll, indicated by the left red arrow in (**C**). (**F**) Black damaged tissue containing fungal mycelium indicated by the right red arrow in (**C**).

**Figure 6 plants-14-02442-f006:**
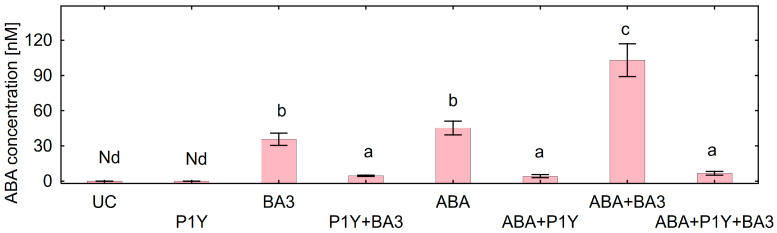
Concentration of abscisic acid (ABA) in the nutrient solution inoculated with *Rhodococcus* sp. P1Y (P1Y) and *Botrytis* sp. BA3 (BA3) and treated with 400 nM of ABA at the end of experiments with sunflower plants. Mean data of three independent experiments. UC stands for un-inoculated control. Nd stands for not detected. Different letters indicate significant differences between treatments (Fisher’s LSD test, *p* < 0.05, n = 9).

**Figure 7 plants-14-02442-f007:**
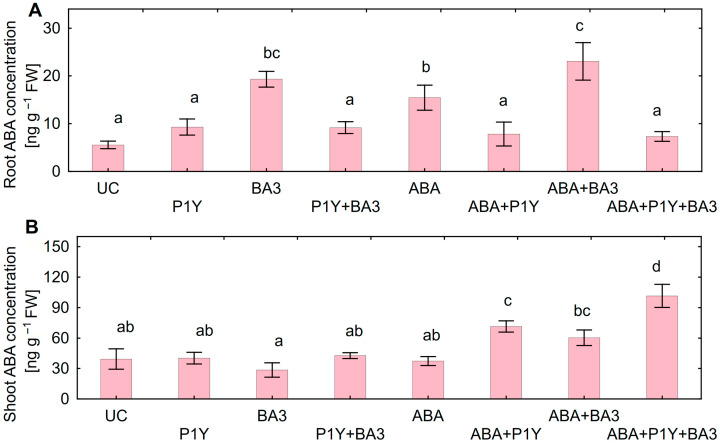
Concentration of abscisic acid (ABA) in roots (**A**) and shoots (**B**) of sunflowers inoculated with *Rhodococcus* sp. P1Y (P1Y) and *Botrytis* sp. BA3 (BA3) and treated with 400 nM of ABA. Mean data of three independent experiments. UC stands for un-inoculated control. Different letters indicate significant differences between treatments (Fisher’s LSD test, *p* < 0.05, n = 9 for roots, and n = 4 for shoots, respectively).

**Figure 8 plants-14-02442-f008:**
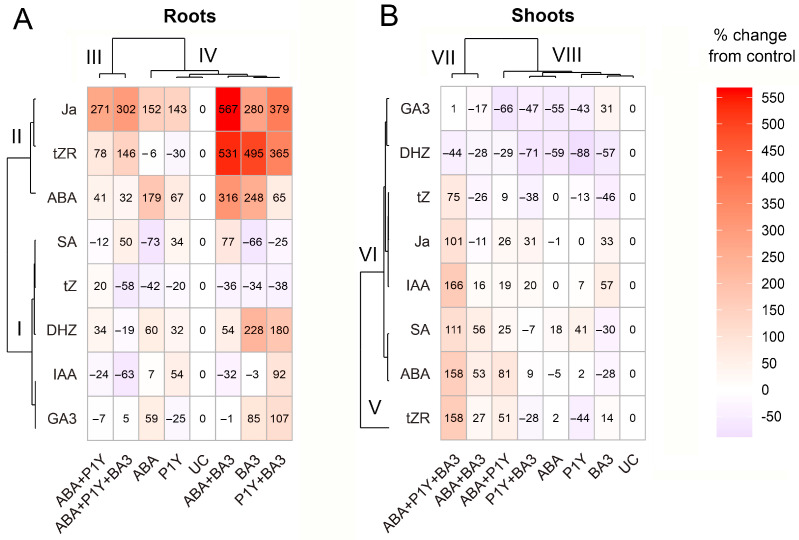
Heat maps of the effects of treatments on phytohormone concentrations and the relationship between phytohormones in roots (**A**) and shoots (**B**). Hierarchical clustering was conducted using Ward’s method (ward.D) based on squared Euclidean distances. Color intensity represents the percent change from control values, where 0% corresponds to the untreated control. Clusters are indicated by the letter C followed by a number.

**Table 1 plants-14-02442-t001:** In vitro production of phytohormones abscisic acid (ABA), indole-3-acetic acid (IAA), indole-3-carboxylic acid (ICA), indole-3-lactic acid (ILA), salycilic acid (SA), jasmonic acid (JA), gibberellic acid (GA3), dihydrozeatin (DHZ), trans-Zeatin (tZ) and trans-Zeatin-ribozide (tZR) by the studied microorganisms (nM).

**Treatments**	**ABA**	**IAA**	**ICA**	**ILA**	**GA3**
*Rhodococcus* sp. P1Y	Nd	40 ± 12	Nd	1660 ± 247	156 ± 72
*Botrytis* sp. BA3	225 ± 23 *	31 ± 12	99 ± 17 *	16,040 ± 3917	1325 ± 165 *
	**SA**	**JA**	**DHZ**	**tZ**	**tZR**
*Rhodococcus* sp. P1Y	Nd	Nd	0.7 ± 0.2	0.8 ± 0.2	Nd
*Botrytis* sp. BA3	8.4 ± 1.2 *	Nd	1.1 ± 0.2	0.9 ± 0.2	Nd

Means and SE of one experiment (n = 2). Nd stands for not detected. Asterisks show significant difference between strains (Student’s *t* test, *p* < 0.05).

**Table 2 plants-14-02442-t002:** Presence of *Rhodococcus* sp. P1Y in the nutrient solution and *Botrytis* sp. BA3 on roots of sunflower seedlings at the end of the experiment.

Treatments	*Rhodococcus* sp. P1Y, 10^6^ CFU mL^−1^	*Botrytis* sp. BA3, µg DNA mg^−1^ root DW
Un-inoculated control	Nd	Nd
*Rhodococcus* sp. P1Y	5.4 ± 0.9 ab	Nd
*Botrytis* sp. BA3	Nd	197 ± 59 a
*Rhodococcus* sp. P1Y + *Botrytis* sp. BA3	5.9 ± 1.1 ab	264 ± 70 a
ABA	Nd	Nd
ABA + *Rhodococcus* sp. P1Y	6.6 ± 0.7 b	Nd
ABA + *Botrytis* sp. BA3	Nd	135 ± 38 a
ABA + *Rhodococcus* sp. P1Y + *Botrytis* sp. BA3	3.9 ± 0.8 a	313 ± 72 a

Data are means ± SE. Different letters indicate significant differences between treatments within columns (Fisher’s LSD test, *p* < 0.05, n = 9 for *Rhodococcus* sp. P1Y, and n = 6 for *Botrytis* sp. BA3, respectively). Nd stands for not detected.

**Table 3 plants-14-02442-t003:** Concentration of indole-3-acetic acid (IAA), salycilic acid (SA), jasmonic acid (JA), gibberellic acid (GA3), dihydrozeatin (DHZ), trans-Zeatin (tZ) and trans-Zeatin-ribozide (tZR) in sunflower plants inoculated with *Rhodococcus* sp. P1Y and *Botrytis* sp. BA3 and treated with 400 µM of ABA (ng g^−1^ FW).

Treatments	IAA	SA	JA	GA3	DHZ	tZ	tZR
Roots							
Un-inoculated control	302 ± 29 b	71 ± 8 cd	4 ± 2 a	2.2 ± 0.3 ab	0.44 ± 0.08 ab	27.9 ± 2.9 cd	2.1 ± 0.2 a
*Rhodococcus* sp. P1Y	467 ± 89 c	95 ± 14 de	9 ± 2 ab	1.7 ± 0.2 a	0.58 ± 0.07 ab	22.3 ± 1.9 bc	1.5 ± 0.3 a
*Botrytis* sp. BA3	294 ± 41 b	24 ± 6 ab	14 ± 1 bc	4.0 ± 0.7 c	1.45 ± 0.21 c	18.6 ± 1.8 ab	12.5 ± 1.8 b
*Rhodococcus* sp. P1Y + *Botrytis* sp. BA3	581 ± 64 c	53 ± 12 bc	18 ± 2 c	4.5 ± 1.3 c	1.23 ± 0.13 c	17.3 ± 1.8 ab	9.7 ± 2.1 b
ABA	323 ± 44 b	19 ± 4 a	10 ± 2 ab	3.5 ± 0.4 bc	0.71 ± 0.08 b	16.3 ± 1.8 ab	2.0 ± 0.3 a
ABA + *Rhodococcus* sp. P1Y	230 ± 26 ab	62 ± 9 c	14 ± 3 bc	2.0 ± 0.2 ab	0.59 ± 0.11 ab	33.6 ± 4.1 d	3.7 ± 0.9 a
ABA + *Botrytis* sp. BA3	206 ± 46 ab	125 ± 21 e	25 ± 4 c	2.2 ± 0.5 ab	0.68 ± 0.12 ab	18.0 ± 3.1 ab	13.2 ± 2.8 b
ABA + *Rhodococcus* sp. P1Y + *Botrytis* sp. BA3	112 ± 16 a	107 ± 14 e	15 ± 2 bc	2.3 ± 0.4 ab	0.35 ± 0.06 a	11.7 ± 0.8 a	5.2 ± 0.6 a
Shoots							
Un-inoculated control	257 ± 41 a	38 ± 4 ab	43 ± 2 a	0.48 ± 0.09 b	0.21 ± 0.04 d	0.91 ± 0.10 ab	0.66 ± 0.10 abc
*Rhodococcus* sp. P1Y	276 ± 21 a	53 ± 4 bc	43 ± 3 a	0.27 ± 0.07 a	0.02 ± 0.01 a	0.80 ± 0.22 ab	0.37 ± 0.06 a
*Botrytis* sp. BA3	404 ± 58 a	26 ± 7 a	57 ± 4 b	0.63 ± 0.13 b	0.09 ± 0.02 abc	0.50 ± 0.19 a	0.75 ± 0.09 abc
*Rhodococcus* sp. P1Y + *Botrytis* sp. BA3	308 ± 38 a	35 ± 3 ab	57 ± 2 b	0.26 ± 0.03 a	0.06 ± 0.02 ab	0.57 ± 0.06 ab	0.48 ± 0.11 ab
ABA	256 ± 33 a	45 ± 7 abc	43 ± 1 a	0.22 ± 0.02 a	0.09 ± 0.02 abc	0.92 ± 0.05 ab	0.67 ± 0.10 abc
ABA + *Rhodococcus* sp. P1Y	306 ± 14 a	47 ± 9 bc	55 ± 1 b	0.16 ± 0.03 a	0.15 ± 0.03 cd	1.00 ± 0.14 b	1.00 ± 0.06 c
ABA + *Botrytis* sp. BA3	298 ± 58 a	59 ± 1 cd	38 ± 1 a	0.40 ± 0.04 b	0.16 ± 0.03 cd	0.68 ± 0.18 ab	0.84 ± 0.09 bc
ABA + *Rhodococcus* sp. P1Y + *Botrytis* sp. BA3	686 ± 97 b	79 ± 9 d	87 ± 8 c	0.49 ± 0.13 b	0.12 ± 0.04 bc	1.60 ± 0.25 c	1.71 ± 0.30 d

Mean data of three independent experiments. UC stands for uninoculated control. Different letters indicate significant differences between treatments in sub-columns for roots and shoots, respectively (Fisher’s LSD test, *p* < 0.05, n = 9 for roots, and n = 4 for shoots, respectively).

## Data Availability

The data presented in this study are available upon request from the corresponding author.

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
