# Peer review of "Abscisic Acid Metabolizing Rhodococcus sp. Counteracts Phytopathogenic Effects of Abscisic Acid Producing Botrytis sp. on Sunflower Seedlings"

_plants, 2025, doi:10.3390/plants14152442_

Round 1

Reviewer 1 Report

Comments and Suggestions for Authors

Shaposhnikov and colleagues submitted a study on the interaction of ABA-metabolizing Rhodococcus sp. with Botrytis sp. on sunflower seedlings under laboratory conditions. The study is of interest for the readership of Plants. However, some issue need to be clarified before publication.

Abstract:

Line 14: PGPR are not necessarly doing biocontrol of phytopathogens. Revise the statement.

Results:

Table1: Only two replicates were tested. This number is far too low to draw any sound conclusion.

Figure 2: change 0,x to 0.x on the y-axis. What are the physiological concentrations of ABA in sunflower plants? Have the authors tried to adapt their concentration to it? Please give rationale in the text.

Figure 3: change 0,x to 0.x on the y-axis. Dry weight data are prefered over Fresh weight data for biomass since the water content can be variably between the samples. Do you have the data? They should be added.

Figure 4:

Why the authors did not use JA-d5 or SA-d5 as internal standard to compensate for variations in ionization efficiency and losses during sample preparation? Please give rationale in the text.

Conclusions: How could the findings be useful in a broader context (practical application)? Please add 1-2 sentences.

It seems to me that the conclusion section would be better placed directly after the discussion. It is a bit difficult to follow when reading first the discussion, then the materials and methods, and then the conclusion.

Materials and Methods:

Fig. S1 looks to me that the plastic cups were covered or sealed with a lid. Eventhough the seedling was surface sterilized inside the seedling microorganism still exist that will interact with the other two ones. I don´t see the advantage of the gnotobiotic conditions. It is not favorable for the plant.

Minor:

Check for uniformity across all units. For example 1.5 mg L-1 not 1.5 mg/L.

Author Response

Dear reviewer!

Thank you very much for taking the time to review this manuscript. Please find the detailed responses below and the corresponding revisions/corrections highlighted/in track changes in the re-submitted files. Please note that the new version of the article indicates the line numbers with responses to your comments, which are those indicated by the Word program.

Comments and Suggestions for Authors

Shaposhnikov and colleagues submitted a study on the interaction of ABA-metabolizing Rhodococcus sp. with Botrytis sp. on sunflower seedlings under laboratory conditions. The study is of interest for the readership of Plants. However, some issue need to be clarified before publication.

Comment 1:

Abstract:

Line 14: PGPR are not necessarily doing biocontrol of phytopathogens. Revise the statement.

Response 1:

Thank you for pointing this out. This sentence revised by adding word “many”.

Comment 2:

Results:

Table1: Only two replicates were tested. This number is far too low to draw any sound conclusion.

Response 2:

We agree that more replicates are needed to obtain a more accurate quantitative assessment. However, with these data we show what hormones the studied strains can produce in principle under favorable conditions in vitro. But, even for such a small number of replicates, for most indicators the standard error of the mean does not exceed 20-25%. Under conditions of interaction with plants, the production of hormones by microorganisms can differ significantly from in vitro conditions. So, we do not draw any important conclusions in the paper based on these quantitative data on ABA concentrations. We were careful not to draw conclusions about the effects of other hormones based on these data. The most important thing was to show that the fungus produces, but the bacteria do not produce, ABA. We have added Student's test data analysis to show the significance of the most discrepant values for phytohormone production by microorganisms. These changes are shown in Table 1 and its footnote.

Comment 3:

Figure 2: change 0,x to 0.x on the y-axis. What are the physiological concentrations of ABA in sunflower plants? Have the authors tried to adapt their concentration to it? Please give rationale in the text.

Response 3:

The comma has been replaced with a dot.

According to literature data, physiological concentrations of ABA in sunflower can vary greatly depending on growing conditions, age and plant organ. Here are some examples: [Dong L, Wu Y, Zhang J, Deng X, Wang T. Transcriptome Analysis Revealed Hormone Pathways and bZIP Genes Responsive to Decapitation in Sunflower. Genes (Basel). 2022 Sep 27;13(10):1737. doi: 10.3390/genes13101737. // Cardoso AA, Brodribb TJ, Kane CN, DaMatta FM, McAdam SAM. Osmotic adjustment and hormonal regulation of stomatal responses to vapour pressure deficit in sunflower. AoB Plants. 2020 Jun 19;12(4):plaa025. doi: 10.1093/aobpla/plaa025. // Mildažienė V, Aleknavičiūtė V, Žūkienė R, Paužaitė G, Naučienė Z, Filatova I, Lyushkevich V, Haimi P, Tamošiūnė I, Baniulis D. Treatment of Common Sunflower (Helianthus annus L.) Seeds with Radio-frequency Electromagnetic Field and Cold Plasma Induces Changes in Seed Phytohormone Balance, Seedling Development and Leaf Protein Expression. Sci Rep. 2019 Apr 23;9(1):6437. doi: 10.1038/s41598-019-42893-5. // Robertson JM, Pharis RP, Huang YY, Reid DM, Yeung EC. Drought-induced increases in abscisic Acid levels in the root apex of sunflower. Plant Physiol. 1985 Dec;79(4):1086-9. doi: 10.1104/pp.79.4.1086].

ABA concentrations in our plants are in this range. Our task was to select a concentration that, on the one hand, was toxic to plants, and on the other hand, was comparable to that produced by the BA3 fungus. Rationale is given in the text on page 5 lines 8-9.

Comment 4:

Figure 3: change 0,x to 0.x on the y-axis. Dry weight data are prefered over Fresh weight data for biomass since the water content can be variably between the samples. Do you have the data? They should be added.

Response 4:

The comma has been replaced with a dot.

We agree with you that DV is a more informative indicator. However, in our case, the plants were grown in a solution in closed vessels, slightly ventilated by means of a filter, which provided high air humidity inside the vessel. Under such conditions, transpiration is not as important as in open greenhouse and field conditions, and the water content is approximately the same for all plants. The plants were taken out of the vessel, weighed and frozen for no more than one minute. Therefore, the dry weight of the plants was not determined.

We are very grateful to you for this remark, since it helped us find a very important typo in Figure 7 and Table 3. The concentration of phytohormones should be indicated per fresh weight of the plants. This error was probably made because we usually express data for vegetation and field experiments exactly in dry weight. These changes were made in Figure 7 and in the text (Page 9, line 21).

Comment 5:

Figure 4:

Why the authors did not use JA-d5 or SA-d5 as internal standard to compensate for variations in ionization efficiency and losses during sample preparation? Please give rationale in the text.

Response 5:

Response: We used high-resolution mass spectrometry with the IT-TOF mass spectrometer (Shimadzu Corporation, Japan) equipped with an ESI interface. Unlike the tqd detector, the masses of all compounds on us instrument are determined with an accuracy of 4 decimal places. The exact gross formula, along with the exact retention time, unambiguously indicate the substance being determined. The method used is described in details in the text (page 16, lines 2-51). The use of high-resolution mass spectrometry allows to select ionization conditions and to study the compounds being determined without the use of expensive labeled compounds.

Comment 6:

Conclusions: How could the findings be useful in a broader context (practical application)? Please add 1-2 sentences.

Response 6:

Such information is added on page 17 line 24.

Comment 7:

It seems to me that the conclusion section would be better placed directly after the discussion. It is a bit difficult to follow when reading first the discussion, then the materials and methods, and then the conclusion.

Response 7:

We used the template recommended by the editors of this journal. The template places Conclusion after Materials and Methods.

Comment 8:

Materials and Methods:

Fig. S1 looks to me that the plastic cups were covered or sealed with a lid. Eventhough the seedling was surface sterilized inside the seedling microorganism still exist that will interact with the other two ones. I don´t see the advantage of the gnotobiotic conditions. It is not favorable for the plant.

Response 8:

We have no evidence that endophytic microorganisms existed in the seedlings we used (page 7 line 12). But gnotobiotic conditions protected the plants from contamination by other microorganisms located outside the pot. Gnotobiotic systems can be very effective in the absence of endophytic microorganisms.

Comment 9:

Minor:

Check for uniformity across all units. For example 1.5 mg L-1 not 1.5 mg/L.

Response 9:

All units were checked for uniformity.

Reviewer 2 Report

Comments and Suggestions for Authors

Dear authors,

The article entitled Abscisic acid metabolizing Rhodococcus sp. counteracts phyto-2 pathogenic effects of abscisic acid producing Botrytis sp. sunflower seedlings addresses a valuable research issue. Still, there are some points to be considered before publication.

  1. One concern is the small number of biological replicates used for quantification in vitro (n = 2, Table 1) and ABA measurements in sunflower shoots (n = 4, Figure 7B). Although the results are shown, the limited sample size reduces the statistical power, and conclusions should be made cautiously. Adding more replicates would enhance data reliability and strengthen the statistical analysis.
  2. While the study convincingly shows that Rhodococcus sp. P1Y decreases ABA levels in the root zone and within plant tissues; the mechanism behind this decrease remains unclear. It would strengthen the paper to provide direct evidence of ABA breakdown in the plant or to clarify whether the reduction occurs through extracellular metabolism or active interaction within root tissues.
  3. The study shows significant changes in plant hormone levels after microbial inoculation. However, it lacks specific controls to determine whether these changes come from microbial hormone production or plant responses to microbes. Using controls like heat-killed microbes, culture filtrates, or isotopic tracing would help identify the source of the hormonal changes.
  4. While the manuscript is scientifically rich and well-referenced, the writing style could benefit from more concise phrasing and clearer paragraph structure. In several sections, particularly the Discussion, the text becomes overly dense, with long paragraphs that combine multiple ideas. I suggest that in order to improve flow, use shorter sentences.

Author Response

Dear reviewer!

Thank you very much for taking the time to review this manuscript. Please find the detailed responses below and the corresponding revisions/corrections highlighted/in track changes in the re-submitted files. Please note that the new version of the article indicates the line numbers with responses to your comments, which are those indicated by the Word program.

Dear authors,

The article entitled Abscisic acid metabolizing Rhodococcus sp. counteracts phyto-2 pathogenic effects of abscisic acid producing Botrytis sp. sunflower seedlings addresses a valuable research issue. Still, there are some points to be considered before publication.

Comment 1:

One concern is the small number of biological replicates used for quantification in vitro (n = 2, Table 1) and ABA measurements in sunflower shoots (n = 4, Figure 7B). Although the results are shown, the limited sample size reduces the statistical power, and conclusions should be made cautiously. Adding more replicates would enhance data reliability and strengthen the statistical analysis.

Response 1:

We agree that more replicates are needed to obtain a more accurate quantitative assessment. However, with these data in Table 1 we show what hormones the studied strains can produce in principle under favorable conditions in vitro. But, even for such a small number of replicates, for most indicators the standard error of the mean does not exceed 20-25%. Under conditions of interaction with plants, the production of hormones by microorganisms can differ significantly from in vitro conditions. So, we do not draw any important conclusions in the paper based on these quantitative data on ABA concentrations. We were careful not to draw conclusions about the effects of other hormones based on these data. The most important thing was to show that the fungus produces, but the bacteria do not produce, ABA. We have added Student's test data analysis to show the significance of the most discrepant values for phytohormone production by microorganisms. These changes are shown in Table 1 and its footnote.

The standard error of the mean for shoot ABA concentrations (n=4) varied from 8% to 25% and the result allowed finding significant differences between means. At the same time no increase in shoot ABA concentrations were detected in single BA3 and ABA treatments. Now we have limited number of samples to make additional biological replicates, but we will take your comment into account in future experiments.

Comment 2:

While the study convincingly shows that Rhodococcus sp. P1Y decreases ABA levels in the root zone and within plant tissues; the mechanism behind this decrease remains unclear. It would strengthen the paper to provide direct evidence of ABA breakdown in the plant or to clarify whether the reduction occurs through extracellular metabolism or active interaction within root tissues.

Response 2:

We agree that studying the mechanism of this effect is very important and interesting task. The studied plant-microbial system is very complex and in-depth studies are required to clarify the mechanisms of interaction between the components. We hope to obtain new information about this in subsequent experiments and think over how to do it. In our opinion, the most probable action of microorganisms is the production and utilization of ABA outside the plant. But microbial ABA, as well as chemical treatment, can penetrate into the roots and cause numerous reactions in plants. We tried to discuss these issues in section 3.4. of the Discussion. However, literary data also indicate the ability of microorganisms that do not utilize or produce ABA to change the concentration of this hormone in plant tissues. We tried not to pay attention to this in the discussion in order to avoid excessive speculation.

Comment 3:

The study shows significant changes in plant hormone levels after microbial inoculation. However, it lacks specific controls to determine whether these changes come from microbial hormone production or plant responses to microbes. Using controls like heat-killed microbes, culture filtrates, or isotopic tracing would help identify the source of the hormonal changes.

Response 3:

Thank you very much for this comment and valuable suggestions. Here we used the treatment with exogenous ABA as a relatively simple additional (specific?) control. It helped to suggest that the effect of BA3 is to some extent was similar to exogenous ABA and fungal ABA could be involved in the effects of the fungus. We continue our research and will take your recommendations into account. We are also working on finding genes for bacterial ABA utilization to generate mutants.

Comment 4:

While the manuscript is scientifically rich and well-referenced, the writing style could benefit from more concise phrasing and clearer paragraph structure. In several sections, particularly the Discussion, the text becomes overly dense, with long paragraphs that combine multiple ideas. I suggest that in order to improve flow, use shorter sentences.

Response 4:

We tried to improve the text and made some sentences shorter. Please find changes on: page 5, lines 8-9; page 10, lines 35-42; page 11 line 26; page 11, lines 46-47.

Round 2

Reviewer 2 Report

Comments and Suggestions for Authors

The revised version addresses the previous concerns. I consider the manuscript suitable for publication.